

# Relationship between *Acropora millepora* juvenile fluorescence and composition of newly established *Symbiodinium* assemblage

Kate M. Quigley[1,2,*], Marie E. Strader[3,4,*] and Mikhail V. Matz[4]

[1] College of Marine and Environmental Sciences, and ARC Centre of Excellence for Coral Reef Studies, James Cook University, Townsville, Queensland, Australia
[2] AIMS@JCU, Australian Institute of Marine Science, Townsville, Queensland, Australia
[3] Department of Ecology, Evolution and Marine Biology, University of California, Santa Barbara, Santa Barbara, CA, United States of America
[4] Department of Integrative Biology, University of Texas at Austin, Austin, TX, United States of America
[*] These authors contributed equally to this work.

## ABSTRACT

Coral-dinoflagellate symbiosis is the key biological interaction enabling existence of modern-type coral reefs, but the mechanisms regulating initial host–symbiont attraction, recognition and symbiont proliferation thus far remain largely unclear. A common reef-building coral, *Acropora millepora,* displays conspicuous fluorescent polymorphism during all phases of its life cycle, due to the differential expression of fluorescent proteins (FPs) of the green fluorescent protein family. In this study, we examine whether fluorescent variation in young coral juveniles exposed to natural sediments is associated with the uptake of disparate *Symbiodinium* assemblages determined using ITS-2 deep sequencing. We found that *Symbiodinium* assemblages varied significantly when redness values varied, specifically in regards to abundances of clades A and C. Whether fluorescence was quantified as a categorical or continuous trait, clade A was found at higher abundances in redder juveniles. These preliminary results suggest juvenile fluorescence may be associated with *Symbiodinium* uptake, potentially acting as either an attractant to ecologically specific types or as a mechanism to modulate the internal light environment to control *Symbiodinium* physiology within the host.

## INTRODUCTION

The establishment of an obligate symbiosis between scleractinian corals and dinoflagellates of the genus *Symbiodinium* is ultimately critical for host survival and tolerance to environmental stressors (*Fitt et al., 2001*; *Hennige et al., 2011*). In species with environmental acquisition of *Symbiodinium* (horizontal transmission), larvae and recently metamorphosed coral hosts are exposed to a diversity of *Symbiodinium*. However critical, few studies have examined mechanisms of *Symbiodinium* uptake and acquisition by the host under natural conditions (*Davy, Allemand & Weis, 2012*). Establishment of

Corresponding authors
Kate M. Quigley,
katemarie.quigley@my.jcu.edu.au
Marie E. Strader,
stradermarie@gmail.com

symbiosis involves initial attraction, uptake of a variable but specific assemblage of *Symbiodinium* followed by winnowing (*Little, Van Oppen & Willis, 2004*; *Rodriguez-Lanetty, Phillips & Weis, 2006*; *Abrego, Van Oppen & Willis, 2009a*; *Yamashita et al., 2014*; *Quigley, Bay & Willis, 2017*). The mechanisms responsible for dinoflagellate-aposymbiotic host contact have been described as broadly chemotactic. However, these studies have been done in a few cnidarian species, including the upside-down jellyfish *Cassiopia xamachana* (*Fitt & Barbara, 1984*), mushroom coral *Fungia scutaria* (*Hagedorn et al., 2015*), and soft coral *Heteroxenia fuscescens* (*Pasternak et al., 2004*) but never in broadcast spawning scleractinians with positively buoyant larvae, extended competency periods and survivorship, as in *Acropora millepora* (*Baird, 2001*; *Graham et al., 2013*). In larvae of *F. scutaria,* trehalose excreted from *Symbiodinium* attracts and initiates feeding behaviour, a potential first step in initiating symbiosis (*Hagedorn et al., 2015*). Adult corals also seed the surrounding sediments with *Symbiodinium*, with an up to 8-fold increase in symbiont abundance in the sediments due to adult seeding (*Nitschke, Davy & Ward, 2015*). Young, negatively buoyant *F. scutaria* larvae are subsequently attracted to adult-seeded sediments (*Hagedorn et al., 2015*). In this scenario, larvae likely recruit and uptake symbionts very close to their parental colonies. However, in species predicted to disperse greater distances, such as *Acropora millepora*, symbiont uptake likely occurs after metamorphosis to ensure association with locally adapted symbiont types (*Howells et al., 2012*), potentially through non-chemosensory mechanisms (*Hagedorn et al., 2015*).

Fluorescence, one of the most conspicuous coral traits, is caused by the expression of fluorescent proteins (FPs). Expression of FPs is strongly regulated in response to environmental perturbations (*D'Angelo et al., 2008*; *Bay et al., 2009*; *Aranda et al., 2011*; *DeSalvo et al., 2012*; *Roth, Fan & Deheyn, 2013*), yet the biological functions of different spectral types of FPs in corals remain unclear. *Acropora millepora* exhibits conspicuous fluorescence polymorphism in larval, juvenile, and adult stages (*Beltran-Ramirez, 2010*; *Kenkel et al., 2011*; *Strader, Aglyamova & Matz, 2018*). A variety of hypotheses regarding the role of FPs on coral performance exist including, but not limited to, FPs acting as photoprotective molecules, dissipating excess light energy in shallow water habitats (*Salih et al., 2000*; *Gittens et al., 2015*), although this hypothesis is not universally supported (*Mazel et al., 2003*; *Roth et al., 2015*). Coral fluorescence interacts with symbiosis in a number of ways. In particular, specific FPs are known to be up-regulated during the initiation of symbiosis (*Voolstra et al., 2009*). The type of symbiont acquired by coral juveniles affects the abundance of green FP: juveniles exposed to non-infective C1 type *Symbiodinium* are significantly more green than juveniles exposed to type D *Symbiodinium* or aposymbiotic juveniles without symbionts (*Yuyama & Higuchi, 2014*). Expression of FPs under thermal stress also varies amongst juveniles hosting different symbiont types. Red and green FP expression increases during thermal stress in juveniles infected with type D symbionts and decreases in those infected with type C1 symbionts (*Yuyama, Watanabe & Takei, 2011*).

Fluorescent proteins can also modulate the internal host light environment by transforming the light spectrum and scattering light (*Salih et al., 2000*). *Symbiodinium* photosynthesis operates most efficiently under blue light in tested *Symbiodinium* types (*Kinzie, Jokiel & York, 1984*) yet *Symbiodinium* exhibit different physiologies under varying
light regimes (*Iglesias-Prieto & Trench, 1994*), depths (*Rowan et al., 1997*; *Bongaerts et al., 2010*) and within coral individuals (*Kemp, Fitt & Schmidt, 2008*). Therefore, it is possible there are varying preferences among *Symbiodinium* types for differences in light intensity available for photosynthesis, which could be modulated by host fluorescence.

Furthermore, while some species of dinoflagellates move in response to light (*Cullen, 1985*; *Horiguchi et al., 1999*), it is unclear whether *Symbiodinium* are able to detect and move toward particular wavelengths emitted by coral juveniles. For example, using differential juvenile fluorescent intensities, signals and patterns for inter/intra-specific communication similar to what has been suggested for corals and reef fish (*Matz, Marshall & Vorobyev, 2006*; *Lagorio, Cordon & Iriel, 2015*). The majority of studies examining fluorescence in corals have targeted the adult stage, with considerably fewer studies examining the role of fluorescence in coral juveniles prior to the onset of symbiosis (*Leutenegger et al., 2007*; *Roth et al., 2007*; *Kenkel et al., 2011*; *Roth, Fan & Deheyn, 2013*; *Strader, Aglyamova & Matz, 2016*). It is currently unclear if corals use variable fluorescence signals that could, for example, allow for the attraction of commensal bacteria or their dinoflagellate symbionts, *Symbiodinium*.

To examine the relationship between coral juvenile fluorescence and the establishment of their symbiotic community, we employed next-generation sequencing of the ITS-2 region to genotype the *Symbiodinium* diversity present in *Acropora millepora* juveniles of varying fluorescence emissions after one month of exposure to sediments. This paper presents evidence for mostly overlapping yet subtly distinct *Symbiodinium* assemblages between green and red juvenile color morphs and discusses potential ecological drivers behind these differences.

## METHODS

### Spawning and larval culturing

Eggs and sperm were acquired from eight gravid *A. millepora* colonies collected from Trunk Reef from about 3 metres depth (AIMS permit number: G12/35236.1) in November 2014 following spawning methods previously described in (*Quigley, Willis & Bay, 2016*). Briefly, positively-buoyant bundles were collected from the colonies, gently mixed with gametes from all 8 colonies, and then left to sit for 1.5 h to allow for fertilization to occur. After this time, fertilized embryos were washed three times by repeated transfer to new fresh-seawater containing bins. Bulk mixtures of gametes were maintained and larvae were raised in tanks with constant aeration and flow through set at 27 °C. Fully competent larvae were exposed to ground, autoclaved crustose coralline algae, a natural known settlement cue (*Heyward & Negri, 1999*), in sterile, plastic 6-well plates and allowed to naturally metamorphose for ~24 h.

### Fluorescence microscropy/ image analysis/ fluorescence quantification

After metamorphosis, fluorescent images of each individual juvenile were taken using a fluorescent stereo-microscope MZ FL-III (Leica, Bannockburn, IL, USA) equipped with a Canon G6 camera. Photos and image analysis was approached as in (*Kenkel et al., 2011*;

*Strader, Davies & Matz, 2015*; *Strader, Aglyamova & Matz, 2016*). Juvenile fluorescence was imaged using the double-bandpass F/R filter (Chroma no. 51004v2), a filter that detects red fluorescence produced from the coral host while excluding any chlorophyll fluorescence. In addition, photographs were imaged prior to treatment exposure to *Symbiodinium*, therefore red fluorescence in coral hosts is not confounded by potential chlorophyll fluorescence. Image analysis was performed using ImageJ (*Schneider, Rasband & Eliceiri, 2012*). For each individual photograph, raw integrated RGB values were calculated across the area of the juvenile. A reference RGB value was calculated within a fixed circular area adjacent to the juvenile in each photograph and was subtracted from measured RGB values for each individual. Individual redness values were calculated as the normalized red value divided by the normalized green value plus the normalized red value.

### *Symbiodinium* acquisition and community sequencing

Post-imaging, the six-well plates with settled juveniles were added to sediment treatments for natural uptake of *Symbiodinium* in the lab in the National Sea Simulator (Seasim) at the Australian Institute of Marine Sciences, resulting in 21 days of cumulative exposure to sediments. Six-well plates with juveniles were floated approximately 5 cm above the sediments using porcelain weights. Flow-through aquaria were fed with 27.2 °C, 0.4 μM filtered seawater that each contained 1L of sediments, with turn-over of the 45L tanks occurring once per hour. During this time, juveniles were exposed to natural light at 50 μmol photon illumination. Given the high level of water filtration and average *Symbiodinium* size of >4 μm (*LaJeunesse et al., 2005*), all *Symbiodinium* taken up by juveniles are assumed to be of sediment origins. Ten and twelve juveniles ($n = 22$) were sampled for green and red color morphs respectively from four tanks (tank 1: three red, three green; tank 2: three red, one green, tank 3: three red, three green, tank 4: three red, three green), and preserved in 100% ethanol and stored at −20 °C until sample processing. DNA extraction and sequence read analysis from individual juveniles followed (*Quigley, Willis & Bay, 2016*). Briefly, nucleic acid isolation included an initial chemical lysis, mechanical lysis using 1 mm silica beads (MPBio, Santa Ana, CA, USA), precipitation and clean-up (*Wilson et al., 2002*). DNA was sent to the University of Texas at Austin's Genomic Sequencing and Analysis Facility (USA) for paired-end Miseq sequencing (Illumina, San Diego, CA, USA) of the ITS-2 region (*Pochon et al., 2001*). Raw reads were cleaned and analysed using the USEARCH and UPARSE pipeline (v.7) and identified to the clade/type level using a custom *Symbiodinium* database constructed from the NCBI database (see Supplementary Information) (*Altschul et al., 1990*; *Camacho et al., 2009*; *Edgar, 2013*).

### Statistical analysis

Cleaned reads were variance normalized to account for differing sequence depth between samples using 'DESeq2' in R (*R Core Team, 2013*; *Love, Huber & Anders, 2014*). The functions 'metaMDS', 'ordiplot', 'ordihull', and 'orditorp' from the 'vegan' package and 'ggplot2' were used to construct NMDS plots using a Bray-Curtis distance matrix on variance-normalized OTU abundance data (*Wickham, 2009*; *Oksanen et al., 2013*). Gradient plots of the smooth response variable "redness" values over NMDS space were constructed using the 'ordisurf' function in 'vegan.'' Discriminant analysis of principle

components (DAPC) was used to identify which OTUs may have attributed to juvenile groupings by fluorescence using the package 'adegenet' (*Jombart, 2008*).

Permutational multivariate analysis of variance using Bray-Curtis distance and Permutation test for homogeneity of multivariate dispersions was used to determine if *Symbiodinium* assemblages differed significantly between red and green juveniles using the 'adonis' and 'betadisper' functions in 'vegan.' To assess if there were significant differences in stabilized abundances between different *Symbiodinium* clades and types between red and green juveniles, independent 2-group Mann–Whitney $U$ Tests (MWU) were performed (at alpha = 0.05). MWU Tests were run using the 'wilcox.test' function in the base R 'stats' package, which are able to account for the lack of homogeneity of variance across samples. To assess if there were significant differences in stabilized abundances between different OTUs found within red and green juveniles, negative binomial generalized linear models in 'DESeq2', using significant Benjamini–Hochberg $p$-values, were performed. The 'ordisurf' function and the 'vegan' package were used to generate Generalized Additive Models to determine if *Symbiodinium* assemblages differed significantly across redness values (*Wood, 2006*; *Wood, 2008*). To test for significant associations between variance normalized abundances and redness values for each OTU, Spearman's rho correlation coefficients and associated $p$-values were calculated per OTU using the 'cor.test' function in the base R 'stats' package. Sample sizes per test for clades and types given in Tables S1 and S2. MWU and Spearman rho correlation tests were also randomized and re-run to assess if significant $p$-values were false-positives.

## RESULTS

### Fluorescent variation among juveniles

We found prominent variation in fluorescent phenotypes among young *A. millepora* juveniles (Fig. 1). Variation in fluorescence, quantified as 'redness', was continuous among individuals, however individuals were also binned into two categories and analysed as either 'red' or 'green' (Figs. 1; 2A, 2B). This fluorescent variation is due to variable expression of *A. millepora* GFP-like proteins, which vary in excitation/emission (*Alieva et al., 2008*; *Beltran-Ramirez, 2010*; *Kenkel et al., 2011*) and copy number (*Gittens et al., 2015*).

### ITS-2 variation between color morphs

Eighty-nine OTUs were recovered between the two color morphs (62 in red and 71 in green). When fluorescence was treated continuously, there was a marginally significant relationship between the *Symbiodinium* community and redness of *A. millepora* juveniles (Generalized Additive Model: $p = 0.053$, $R^2 = 0.33$) (Fig. 2A). Of the 89 OTUs retrieved across juveniles, Spearman's Rank Correlation tests indicated significant candidate OTUs as correlated with redness as a continuous variable (OTU79-C15, OTU985-A1; Fig. S1A). However, randomization tests (where redness values were shuffled among samples) indicated that these relationships might be false positives.

There was no significant relationship between *Symbiodinium* assemblage composition and fluorescence when color was treated as a categorical trait as found using either permutational manova (Adonis with Bray-Curtis distance: $p = 0.58$) (Fig. 2B) or

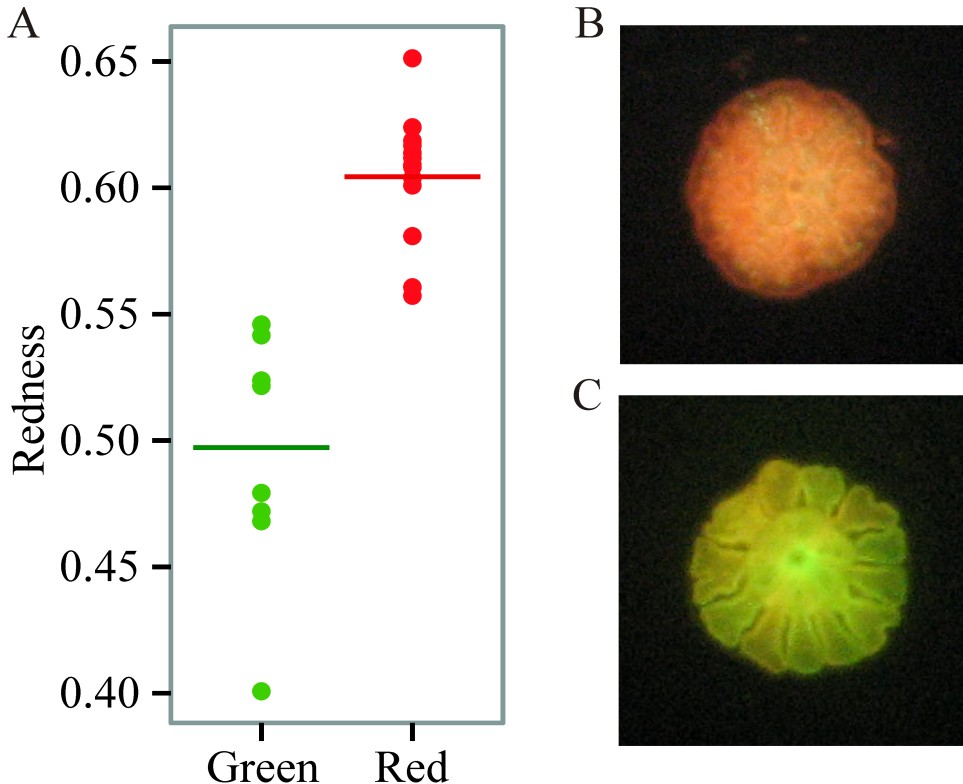

**Figure 1** **Variation in juvenile fluorescence.** (A) Fluorescence as a categorical trait ($x$-axis) and as a continuous trait ($y$- axis). (B) A red morph. (C) A green morph.

discriminant analysis (Fig. 2C). OTU427 (C15) contributed the most to differentiating between the assemblages in the two color morphs as shown through loading weights of each OTU above discriminant analysis default thresholds (Figs. 2B, 2C). Although it appeared that green juveniles displayed a greater capacity to associate with a wider range of *Symbiodinium* OTUs and red juveniles displayed a more restricted assemblage (Figs. 2B, 2C), there was no significant relationship between the variances of the two color morphs (Permutation test for homogeneity of multivariate dispersions, $df = 1$, $p = 0.5$).

Clade and type abundances varied between color morphs (Fig. S1B), although after randomization, only clade A was robustly found in significantly greater abundances in red juveniles (Mann–Whitney $U$ Tests, $p = 0.03$). The same OTU (427-C15) predicted as the OTU driving the greatest divergence of the two communities in the discriminant analysis was found at 15 fold lower abundances in red compared to green juveniles and OTU985 (*S. microadriaticum*) occurred in 74 fold higher abundance in red compared to green juveniles (DESeq2-Bejamini-Hochberg p-adjusted values both = 0.03).

## DISCUSSION

This study provides the first evidence of the potential role of fluorescence on the establishment of symbiosis in a broadcast spawning, horizontally transmitting reef-building

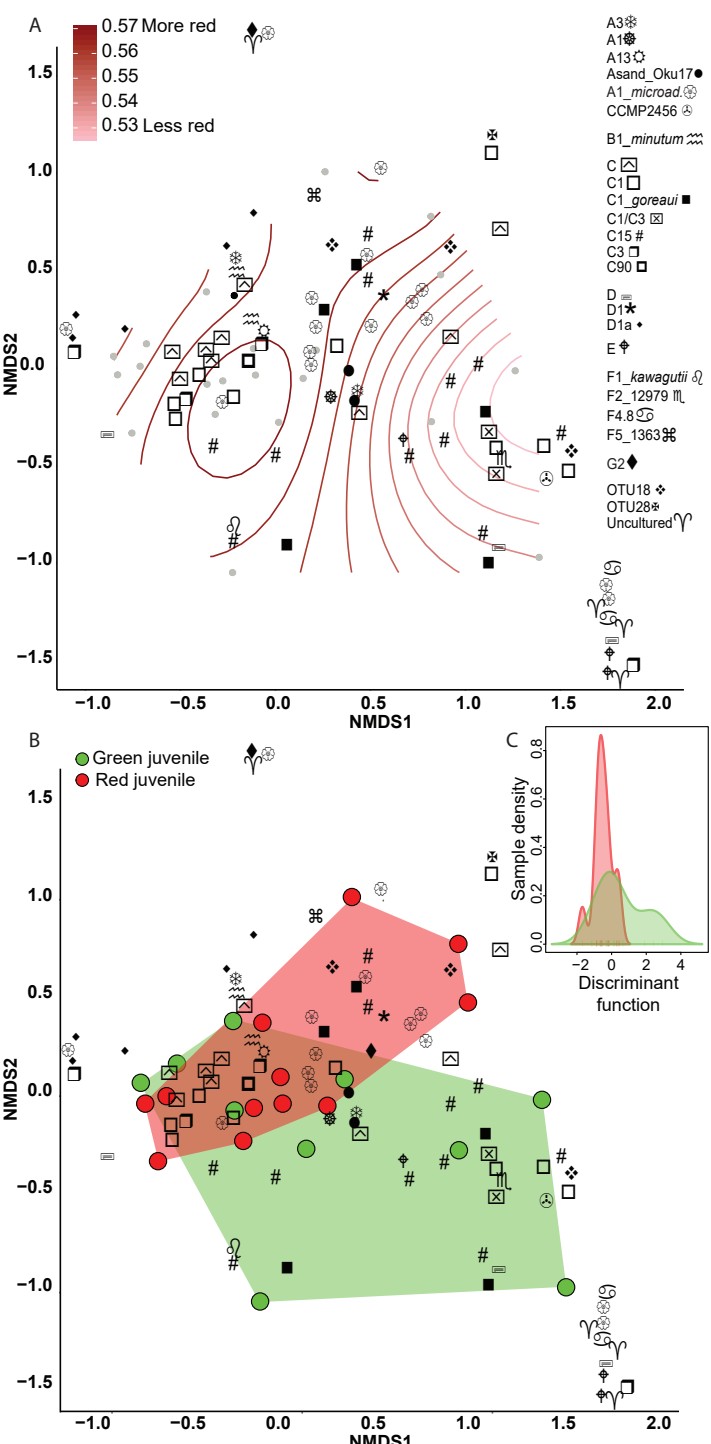

**Figure 2** **Non-metric multidimensional scaling (NMDS) using Bray Curtis distance matrix of variance normalized *Symbiodinium* abundances.** (A) Fluorescence as a categorical variable (red or green), and (B) as a continuous variable (redness). (C) Discriminant analysis of principle components (DAPC) with fluorescence as a categorical variable. Symbol shapes correspond to clade designations with multiple types (A, circular shapes; C, squares; D, star/varied shapes; F, line based shapes; Uncultured/OTUs, cross shapes).

coral. We find that *Symbiodinium* assemblages mostly overlap among juvenile fluorescent morphs with marginally significant separation between them when fluorescence is analysed as a continuous trait, driven mostly by variation in A1 and C15. These results emphasize that any future studies need to account for the continuous nature of fluorescent variation among recruits, rather than binning individuals into separate color categories. Despite our findings, there are substantial caveats in this study, notably the heritability of fluorescence in larvae of this species (*Kenkel et al., 2011*) and the fact that our experimental design did not disentangle potential parental effects. While our study is correlative, it opens the door for future studies to identify the causes of the relationship between juvenile fluorescent color and the initial *Symbiodinium* community.

## Variable *Symbiodinium* types between color morphs

We detected significant changes in the *Symbiodinium* community as redness values shifted, potentially driven by significantly higher abundance of clade A in redder individuals and clade C in greener juveniles. Clade A is a generalist clade able to tolerate low and high light levels (*Iglesias-Prieto & Trench, 1997a*) and exhibits decreased growth in red compared to green light (*Kinzie, Jokiel & York, 1984*). C15 *Symbiodinium* often inhabit extreme shallow environments of high irradiance in species such as *Montipora digitata*, but can also dominate across substantial geography and depth in *Porites spp.* (*Veron, 2000*; *LaJeunesse et al., 2003*; *LaJeunesse et al., 2004*). It is possible that different abundances of specific *Symbiodinium* types in red and green fluorescent juveniles and significant changes to assembly composition with redness was observed because variation in the host internal light environment, or the availability of photosynthetically usable light, affects the proliferation or physiology of various *Symbiodinium* types differently. Spectral properties in coral tissues are regulated by FPs, with different FP types significantly changing the available wavelengths of light inside coral tissues (*Salih et al., 2000*). Green FP absorbs blue light and therefore diminishes the light available for photosynthesis and can suppress the cell-cycle of *Symbiodinium* (*Kinzie, Jokiel & York, 1984*; *Alieva et al., 2008*; *Wang et al., 2008*; *Suggett et al., 2015*). Therefore, GFP might act to regulate the amount of photosynthetically available light or proliferation of *Symbiodinium*, thus allowing uptake of an assemblage with specific physiologies. Alternatively, more red individuals are unable to regulate photosynthetically available light for *Symbiodinium*, which may promote the higher abundance of generalist clade A *Symbiodinium* and might reflect uptake of a more neutral *Symbiodinium* assemblage.

In addition, it is possible that greener host individuals are better at attracting more evolutionary derived C15 *Symbiodinium* as a way to avoid more opportunistic clade A *Symbiodinium*. Furthermore, opportunistic clade A *Symbiodinium* could avoid greener individuals as a means of increasing a competitive advantage with other *Symbiodinium* types. However, the actual dynamics of host/symbiont control regarding uptake and winnowing needs to be determined with follow-up studies. Regardless of what mechanisms may be at play, our results suggest subtle differences in *Symbiodinium* assemblages along a range of recruit fluorescence and drivers of these differences have yet to be investigated. It is also possible that the differences we observe may reflect variable irradiance preferences

between *Symbiodinium* for specific light environments, although this is highly speculative and necessitates further study.

Differences in irradiance preferences may vary across specific *Symbiodinium* strains (*Rowan et al., 1997*), which could reflect differences in photo-acclimation potential. It is unclear how difference in irradiance preferences could impact coral performance, although perhaps this could occur through variability in carbon/nitrogen acquisition across strains, which occurs along depth gradients in *Stylophora pistillata* (*Ezzat et al., 2017*). Intra-clade level differences are extensive across a range of *Symbiodinium* traits and therefore the clade level may not be a good predictor of physiology. For example, in a comparative study, *Symbiodinium microadriaticum* and *Symbiodinium pilosum* (A2) had distinct photophysiological characteristics (*Iglesias-Prieto & Trench, 1997b*), and type A3 was found living in high irradiance environments through the production of MAA mycosporine-glycine (*Banaszak et al., 2006*). Photophysiological traits such as electron transport rates, cellular RCII concentrations, and light harvesting also differed across distinct types and did not cluster by clade; again suggesting that types within clades A and F are physiologically distinct (*Iglesias-Prieto & Trench, 1997a*; *Suggett et al., 2015*). Finally, the unique OTUs for both color morphs suggests that these are distinct subtypes and that variations between these types exist (for example: C3-u vs. C3-z prevalence between light environments (*LaJeunesse et al., 2010*).

It is possible that other physiological differences among juveniles besides fluorescence may be influencing *Symbiodinium* uptake. The higher abundances of specific *Symbiodinium* types may be due to different by-products produced by these two juvenile color morphs. The sugar trehalose has been found to be a strong chemical attractant for coral larvae (*Hagedorn et al., 2015*) and different *Symbiodinium* types may be similarly attracted to trehalose or other sugar derived chemicals. Potentially, different color morphs could produce variable attractive compounds or quantities. Indeed, the variable production of N-acetyl glucosamine and C6 sugars in coral mucus has been found to significantly alter bacterial assemblages (*Lee et al., 2016*). Finally, since our experiment was performed on a bulk culture containing multiple genotypes, it is also possible that the variation we see in regards to *Symbiodinium* could be a result of genotypic variation influencing uptake as there are heritable components to both fluorescence and *Symbiodinium* acquisition (*Kenkel et al., 2011*; *Quigley, Willis & Bay, 2017*). Further work is needed to determine if other traits aside from variable fluorescence are associated with juvenile physiology that may explain variable uptake of symbionts.

Although we found a significant correlation between the abundance of two specific OTUs representing two *Symbiodinium* types and redness fluorescence, a previous report found no significant relationship between symbiont abundance and green fluorescence in *Seriatopora hystrix* (*Roth, Fan & Deheyn, 2013*). This may be due to three reasons: (1) that study did not differentiate between different *Symbiodinium* clades or types, (2) all *Seriatopora hystrix* larvae measured were only of green fluorescence and did not exhibit red to green variability as measured in *A. millepora,* and (3) differing methods in *Symbiodinium* quantification. In addition, it is possible that fluorescence shifts during juvenile ontogeny, making comparisons between larval and juvenile fluorescence incongruent.

### Outlook for future research

Despite the caveats in our results, this study highlights the potential of juvenile fluorescence to modulate the host light environment impacting the physiology of specific *Symbiodinium* types or function as an attractant for specific types of ecologically diverse *Symbiodinium*. Any studies going forward would need to incorporate larger sample sizes, as it is possible that the low number of sequenced animals in this study limited our power to distinguish between significant changes and noise in *Symbiodinium* types and OTUs variable between fluorescent morphs. The first step forward would be to disentangle potential parental effects of fluorescence and *Symbiodinium* assemblage uptake (*Kenkel et al., 2011*; *Quigley, Willis & Bay, 2017*). Despite potentially confounding parental effects, our study provides the first evidence that juvenile fluorescence is a trait potentially linked to mechanisms of *Symbiodinium* acquisition and warrants further investigation. In addition, it would be ideal to monitor the time course of symbiont winnowing among fluorescent color morphs, as the study presented here is merely a snapshot in time during a dynamic period of restructuring of *Symbiodinium* assemblages (*Quigley, Bay & Willis, 2017*). It is possible that the differences in *Symbiodinium* OTUs and types we observe in this study are due to differences in the timing of winnowing between color morphs (*Abrego, Van Oppen & Willis, 2009b*). If true, this would support a role of juvenile fluorescence in the winnowing process. It would also be ideal to disentangle if *Symbiodinium* acquisition is dependent on the light transforming properties of FPs by performing manipulative light experiments using specific filters to modulate the host fluorescence available to *Symbiodinium* prior to infection, as in (*Strader, Davies & Matz, 2015*). This would test the hypothesis of host fluorescence acting as an attractant, a hypothesis that has also been proposed in (*Hollingsworth et al., 2005*). It is imperative to investigate the spectral and behavioural properties of specific symbiont types, specifically characterizing the wavelengths of light that can be detected as well as irradiance preferences between specific *Symbiodinium* types (*Suggett et al., 2015*; *Innis et al., 2018*). For example, our data suggest that clade A could be less attracted to high abundances of GFP in juveniles, although the physiological response of clade A *Symbiodinium* to different light spectra and intensity need further substantiation. To take it a step further, it would be interesting to identify if different irradiance preferences were related to differential nutrient acquisition or other aspects of coral performance (*Ezzat et al., 2017*). Finally, our findings provide a framework for further experimental studies exploring the ecological function of fluorescence during coral ontogeny and its potential influence on the establishment of symbiosis.

## CONCLUSIONS

To the best of our knowledge, this is the first study to examine the potential function of coral fluorescent proteins in modulating the uptake of *Symbiodinium* in broadcast spawning corals with horizontal symbiont acquisition. We find that in *A. millepora*, juvenile fluorescence varies continuously between green and red and this fluorescent variation is associated with preferential uptake of clade A in redder individuals and type C15 *Symbiodinium* in greener individuals. The biological significance of these associations has

yet to be determined; however; we hope this work will provide a platform for future studies investigating the functional significance of the association between coral fluorescence and initial uptake of *Symbiodinium* assemblages.

### Funding

Funding was provided by the Australian Research Council through ARC CE1401000020. The funders had no role in study design, data collection and analysis, decision to publish, or preparation of the manuscript.

### Grant Disclosures

The following grant information was disclosed by the authors:
Australian Research Council: ARC CE1401000020.

### Competing Interests

The authors declare there are no competing interests.

### Author Contributions

- Kate M. Quigley conceived and designed the experiments, performed the experiments, analyzed the data, contributed reagents/materials/analysis tools, prepared figures and/or tables, authored or reviewed drafts of the paper, approved the final draft.
- Marie E. Strader conceived and designed the experiments, performed the experiments, contributed reagents/materials/analysis tools, prepared figures and/or tables, authored or reviewed drafts of the paper, approved the final draft.
- Mikhail V. Matz contributed reagents/materials/analysis tools, authored or reviewed drafts of the paper, approved the final draft.

### Field Study Permissions

The following information was supplied relating to field study approvals (i.e., approving body and any reference numbers):

Field experiments were approved by the Great Barrier Reef Marine Park Authority (GBRMPA) (Permit Number G12/35236.1).

### Data Availability

Sequences are available using the SRA accession: SRP133664.

### Supplemental Information

Supplemental information for this article can be found online at http://dx.doi.org/10.7717/peerj.5022#supplemental-information.

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
