# Peer review of "Relationship between Acropora millepora juvenile fluorescence and composition of newly established Symbiodinium assemblage"

_PeerJ, doi:10.7717/peerj.5022_

## Round 0.1 · original submission · Minor Revisions

Your manuscript has now been reviewed by two researchers with extensive knowledge of the subject area of zooxanthella fluorescence and acquisition in hermatypic corals. Both praised the clarity of your manuscript and provided clear, focused recommendations for improvement. While both have recommended "minor revisions" I consider their recommendations to be of high quality and all should be carefully considered and either adopted or not with clear explanation. Congratulations on the excellent reviews.

·

Basic reporting

This study provides evidence for the role of coral fluorescence in affecting symbiont assemblages in early life stages of a recently metamorphosed aposymbiotic reef coral. While their results are preliminary, the results are interesting and informative and provide a framework that motivates further research. I find the manuscript well written with appropriate scholarship. My comments mostly consist of grammatical or stylistic suggestions or requests for clarity.

Abstract
Line 37: delete extra “as” here, in “acting as either as an…”

Introduction
The introduction paints a good picture of the literature and knowledge gaps, however, is there any room for the authors to address (perhaps briefly at Line 70) the role host FPs play during in bleached corals/during thermal stress?

Line 48: appears to be an extra space after “symbiosis”

Line 57: can the authors provide a citation for greater dispersal potential in brooded larvae

Line 93: recommend breaking this sentence in two for clarity, perhaps with period before “for example”.

Line 104: recommend incorporating “through variable fluorescent signal” above at Line 102. Feels redundant if retained at both Line 102 and 104.

Line 106: I commend the authors on this straightforward and pithy culmination to the introduction.

Experimental design

The largest issue I have is there is little information on the experimental design at the treatment level (see comment below). I would like the authors to address this concern and provide detail on the bulk culturing/sedimentation treatment and the conditions the juvenile corals were maintained under during the 21d period.

Paragraph (2. Fluorescence…)
I do not have an issue with how color scores were determined, however, considering there are many published examples of color scoring, I feel a citation in necessary for at least some portion of this approach. Can the authors provide such a citation?

Line 115: what depth were these colonies at? Could depth of parent affect the properties of offspring here—perhaps the parental effects your elude to in discussion.

Line 129: prior to exposure to sediment treatments but perhaps not wholly aposymbiotic? Could a few words be added here to clarify that ‘exposure’ references treatment exposure but not in situ exposure?

Line 138: there is limited information on sediment treatments consisted of. What were the conditions during these sediment treatments—ie., light, temp, flow, aquaria or well plates, in situ or in the lab? Were all symbionts assumed to originate in sediment or in seawater, was this natural seawater or filtered, were there statistical replicate treatments or a common garden…. Please clarify and provide more information on your treatment.

Line 170: errant “of” here at “…of OTUs”

Validity of the findings

The authors have interesting findings and are quite parsimonious in their interpretation—ie, testing for false positives and the discussion of correlation in lines 227. I appreciate the conservative tone and agree this study provides needed insights for a larger discussion on the functional roles of the continuous distribution of the color and fluorescence properties of corals.

RESULTS
Line 204: need to make “Discriminant” lowercase

DISCUSSION
Line 221: recommend adding broadcast spawning in this description, since ‘highly dispersive’ is a not very specific

Line 239: perhaps mention that C15 can also dominate across substantial geography and depths in specific genera such as Porites (LaJeunesse, et al. (2004). High diversity and host specificity observed among symbiotic dinoflagellates in reef coral communities from Hawaii. Coral Reefs.)

Line 246: suggest “suppress” as opposed to retard

Line 258-269: could this ‘preference’ also be referred to as photo-acclimatization potential? Is there evidence the authors could cite to clarify this point—I recognize it is speculative, but some clarity in the niche partitioning vs. acclimatization capacity would be informative. i.e., how can this ‘preference’ influence coral performance? Suggest,
Ezzat, et al (2017). Carbon and Nitrogen Acquisition in Shallow and Deep Holobionts of the Scleractinian Coral S. pistillata. Frontiers in Marine Science, 4, 2273–12Line 261: here and elsewhere where species names for Symbiodinium types are referenced, I think you need to give full species name (Symbiodinium and not “S”) since these full specific epithets have not yet been introduced.

Line 269: missing a closing parenthesis here

Line 286: in which species? S. hystrix?

Line 279: this ‘bulk culture’ information needs to be detailed in the methods. It is unclear how this was performed.

Line 309-320: I agree with the authors here, and see a lot of opportunity to clarify these relationships in future studies. This is a very well constructed paragraph.

Line 319: Suggestion: Innis, et al. (2018). Coral color and depth drive symbiosis ecology of Montipora capitata in Kāne‘ohe Bay, O‘ahu, Hawai‘i. Coral Reefs, 1–8

Additional comments

A general comment: Was there any attempt to measure fluorescence at the end of the experiment? Could the same chlorophyll fluorescence subtraction (Line 128) have been done to determine if host fluorescence was maintained over time or during winnowing (as discussed in Line 307)? Could the authors provide any information on whether fluorescence may be fixed or variable during concurrently with ontogeny and symbiont winnowing?

Reviewer 2 ·

Basic reporting

This manuscript was clear and easy to understand. The literature background/context was well supported, but see my comment below regarding a couple of additional citations I suggest including. The article structure, figures, and tables are professional, and raw data is available via the NCBI Sequence Read Archive. I did notice, however, that in the specified SRA entry, there were only 18 samples, compared to the 22 samples referenced in the text. Can you please clarify this difference?

Literature/Citation Suggestions:
I recommend mentioning the “Beacon hypothesis” by Hollingsworth et al (2005, Coral Reefs; 2006, Proceedings of the 10th ICRS) which demonstrated that motile Symbiodinium exhibit phototaxis towards green light. I think this should be discussed at least briefly, potentially when you mention that “This would test the hypothesis of host fluorescence acting as an attractant” (Lines 316-318), as it is directly relevant here.
You have done an excellent job of supporting your conclusions as well as proposing alternate mechanisms that may drive the patterns you observed. I will suggest one more alternate hypothesis – it is possible that the “green” juveniles are better at attracting more evolutionarily derived symbionts (i.e. C15), as the C15s may hone in on the brighter GFP, and be taken up by the coral. Since infection is attraction plus selective uptake (Yamashita et al 2014), perhaps those corals then do not need/want to uptake clade A Symbiodinium. It is also possible that “opportunistic” As stay away from bright green juveniles, and instead prefer the redder individuals (as you mention, because there is more light, and may be a better environment for them to compete with other Symbiodinium types/clades).

Experimental design

The research question was well defined and relevant. I was excited to see the authors investigating how juvenile fluorescence relates to symbiont uptake, as I believe it addresses an intriguing gap in our current knowledge. This manuscript and its conclusions would be strengthened by inclusion of parental information (i.e. relatedness of coral juveniles), but I acknowledge that is probably not possible at this point, and I think the authors did a good job of discussing this, proposing alternative hypotheses, and setting the stage for future research.
NMDS is good for visualizing multidimensional data, but I would like to see what the output of a constrained ordination looks like for these data. Specifically, you could use capscale in vegan to run this analysis with a call something like ord <- capscale(SymbioCommunity ~ redness) followed by anova(ord). Is this model significant?
Is the custom Symbiodinium database that you used (line 148) available somewhere for download? If so, please provide a citation for the database (I think the current citations are just for the NCBI database in general). If not, please include the database either with your code, and/or on GitHub, and/or somewhere else online. This will help with reproducibility of your study.

Validity of the findings

The findings of this study are valid, well supported, and alternative hypotheses are clearly presented.

Additional comments

Overall, I think this is an interesting manuscript that is worth publishing. I have included several comments below with specific suggestions for making the figures clearer and more intuitive.
Figures:
Fig 2a. The contour coloring would be more clear and consistent if the color bar went from red to green. This is because the color differences on the figure are a bit hard to distinguish. If you set a mid-line of 0.55 (as in Fig 1), with green (low) to red (high) this may be clearer. (Although check that it prints ok in black and white before making this change).
I find the use of randomly-chosen symbols for the different Symbiodinium types to be distracting. I acknowledge that this may be an inherent problem with so many types, but is it possible to try to make the symbols for each type within a clade, similar? For example, most of the Cs are somewhat box-shaped, except “C” and “C1*”, but D1 and Uncultured are also box-shaped. Perhaps you could switch those, so all Cs are box-shaped? I realize that this may seem picky, but as it is, it is fairly difficult to see the broader-scale patterns.
Fig 2b. For black-and-white printing, I’d suggest changing the shape of one of the juveniles, so they’re not distinguished only by color. I like that they are distinguished by color, but you could make one of them a box or a triangle for clarity.
Figure S1a. Both of these sub-figures should be truncated at 0 (i.e. the fitted model and confidence intervals shouldn’t cross 0, as a negative mean abundance isn’t possible). It also looks like the mid-line (yellow, between green and red) for your color bar is at 0.5. If possible, please change to 0.55, so it matches Fig 1.
Fig S1b. If you move up the second bar from the left on the bottom panel to the panel above it, you could expand the axes for the bottom panel (e.g. from 0 to 20) which would make the very low abundance symbionts easier to see.

---

## Round 0.2 · accepted · Accept

I very much appreciate the clear and thorough responses and alterations made in consideration of the reviewer recommendations. An editor's dream!

#